# Free ISG15 inhibits Pseudorabies virus infection by positively regulating type I IFN signaling

**Huimin Liu**[1]☯, **Chen Li**[1]☯, **Wenfeng He**[1]☯, **Jing Chen**[1], **Guoqing Yang**[1], **Lu Chen**[2]*, **Hongtao Chang**[2]*

**1** College of Life Sciences, Henan Agricultural University, Zhengzhou, Henan, China, **2** College of Veterinary Medicine, Henan Agricultural University, Zhengzhou, Henan, China

☯ These authors contributed equally to this work.
* chenluhau@126.com (LC); ndcht@163.com (HC)

**Data Availability Statement:** The data underlying the results presented in the study are available from.

**Funding:** This work was supported by the National Natural Science Foundation of China (31902268 to

## Abstract

Interferon-stimulated gene 15 (ISG15) is strongly upregulated during viral infections and exerts pro-viral or antiviral actions. While many viruses combat host antiviral defenses by limiting ISG expression, PRV infection notably increases expression of ISG15. However, studies on the viral strategies to regulate ISG15-mediated antiviral responses are limited. Here, we demonstrate that PRV-induced free ISG15 and conjugated proteins accumulation require viral gene expression. Conjugation inhibition assays showed that ISG15 imposes its antiviral effects via unconjugated (free) ISG15 and restricts the viral release. Knockout of ISG15 in PK15 cells interferes with IFN-β production by blocking IRF3 activation and promotes PRV replication. Mechanistically, ISG15 facilitates IFNα-mediated antiviral activity against PRV by accelerating the activation and nuclear translocation of STAT1 and STAT2. Furthermore, ISG15 facilitated STAT1/STAT2/IRF9 (ISGF3) formation and ISGF3-induced IFN-stimulated response elements (ISRE) activity for efficient gene transcription by directly interacting with STAT2. Significantly, ISG15 knockout mice displayed enhanced susceptibility to PRV, as evidenced by increased mortality and viral loads, as well as more severe pathology caused by excessive production of the inflammatory cytokines. Our studies establish the importance of free ISG15 in IFNα-induced antiviral immunity and in the control of viral infections.

## Author summary

By antagonizing type I IFN production and action, pseudorabies virus (PRV) evades host defense to establish persistent infection. Interferon-stimulated gene 15 (ISG15) is known to inhibit the replication of many viruses, although pro-viral effects of ISGylation are also reported. We previously found that ISG15 was strongly induced upon PRV infection, however, how ISG15 abundance is regulated and whether the ISG15 influences PRV replication via monomer or polypeptide remain unclear. This study shows that ISG15 restricts PRV growth dependent on free ISG15, moreover, free ISG15 positively regulated IFNα-

H.L. and 31772781 to L.C.) and the Youth Backbone Teachers' Training Program of Colleges and Universities of Henan Province (2021GGJS034 to H.L.). The funders had no role in study design, data collection and analysis, decision to publish, or preparation of the manuscript.

**Competing interests:** The authors have declared that no competing interests exist.

mediated antiviral activity by facilitating activation and nuclear translocation of STAT1 and STAT2. Furthermore, ISG15 knockout mice display more highly sensitive to PRV, indicating that ISG15 may also have an antiviral function *in vivo*. This study highlights that free ISG15 is a critical antiviral target against PRV infection and improves our understanding of the host immune response to PRV infection.

## Introduction

Pseudorabies virus (PRV), also called *suid* herpesvirus 1 (SuHV-1) or *Aujeszky's disease* virus (ADV), belongs to the alphaherpesvirus subfamily and infects a broad host range including its natural host swine [1]. Particularly, recent evidence revealed that PRV can induce serious encephalitis in a small portion of the infected individuals, raising the concern of PRV cross-species transmission [2–6]. Like other herpesvirus, PRV can establish a latent infection in peripheral nerve cells, which is usually used as a model for studying the biology of alphaherpesvirus [7,8]. Despite intensive research, neither viable therapeutic options nor effective vaccines is currently available to prevent PRV infection [2,9]. Therefore, understanding the interplay between PRV and host cells will improve antiviral treatment.

In response to viral invasion, the host evolves various defense mechanisms. Among these, the type I interferon (IFN-I) plays a critical role in host innate immunity defense against viral infection. IFN-I, represented by IFNα/β, binds to their respective receptors and activates the JAKs, which subsequently phosphorylate STAT1 and STAT2. The phosphorylated (p-) STAT1 and p-STAT2 complex with IRF9, resulting in the formation of ISG factor 3 (ISGF3). ISGF3 shuttles to the nucleus, where it binds to the IFN-stimulated response element (ISRE) in DNA and stimulates the transcription of hundreds of interferon-stimulated genes (ISGs) involved in antiviral immune responses [10,11]. Increasing evidence indicates that PRV utilizes its encoded proteins to antagonize the IFN response by suppressing IFN-I signaling, or blocking IFN downstream ISGs expression [12–14].

Interferon-stimulated gene 15 (ISG15) is an IFNα/β-induced ubiquitin-like protein that exists in two distinct states: as a free molecule (intracellular and extracellular) or covalent conjugation to lysine residues of target proteins (ISGylation). Similar to ubiquitination, ISGylation involves a cascading reaction catalyzed by E1 activating (UbE1L), E2 conjugating (UbcH8) and E3 ligase (Herc5) enzymes, which are also induced by IFNα/β [15]. ISG15 can be removed from its target proteins by the ubiquitin-specific protease USP18, making the ISGylation process reversible [16–19]. Several studies have suggested a role for ISG15 has proviral or antiviral activities, depending on the virus and host species [20,21]. However, the role of ISG15 in viral infection remains controversial [22]. In vitro studies in mouse cells have demonstrated an antiviral role for ISG15 during several viral infection [23–25], although there are some reports of viruses displaying no enhanced replication when ISG15 is deficient [26,27]. Knocking down ISG15 in human cells has also suggested an antiviral role for ISG15 during infection with numerous viruses [23,28–30], while other studies have suggested no role at all [31,32]. Furthermore, mice lacking ISG15 exhibit enhanced susceptibility to some but not all viruses [26,27,33–35], while ISG15 deficiency in human enhanced viral resistance [22,36–38]. More recently, ISG15 was reported to have an immunomodulatory effect by acting as a negative regulator of IFN-I signaling, thus regulating the antiviral response during viral infection [39].

ISG15 is strongly upregulated in porcine kidney epithelial cells (PK15) following PRV infection [40]; however, its regulation during infection and the role of ISG15 in viral growth have not been characterized. In this study, we show that ISG15 expression and ISGylation are initially induced after PRV infection but later suppressed by viral responses, and that gE, plays an

important role in reducing ISG15 expression. By silencing the expression of ISG15, we show that ISG15 inhibits PRV replication via the free ISG15, which promotes IFN-β production to suppress viral growth. Moreover, we reveal that ISG15 silencing impairs IFN$\alpha$-mediated anti-PRV effect by blocking phosphorylation and nuclear translocation of STAT1 and STAT2. Significantly, ISG15 knockout mice exhibited highly susceptible to PRV infection, as evidenced by high mortality, increased viral titer and more severe inflammatory. Our results reveal a critical role for ISG15 in IFN$\alpha$-mediated antiviral activity and will provide a potential cellular therapeutic strategy.

## Results

### Free ISG15 and conjugated protein accumulation during PRV infection

Although ISG15 expression has reportedly increased during PRV infection in our previous study, how free ISG15 and ISG15 conjugates impact PRV growth and their roles in host defenses against PRV remain largely unknown. We sought to dissect the expression of free ISG15 and ISG15 conjugates during PRV infection with different multiplicity of infections (MOIs) and different times post-infection. In PRV-infected PK15 cells, the greater levels of free ISG15 and ISG15 conjugates were observed at 24 hours post-infection (hpi), even at relatively low MOIs (0.5, 1 and 5). However, the levels of free ISG15 and ISG15 conjugates at high MOI (10) were much lower than those at low MOIs (Fig 1A–1C), and the decline in ISG15 expression was correlated with increased abundance of a representative viral late protein, gE, by 12 h and 24 h. This suggested that the high levels of ISG15 present in the cells to prevent PRV growth, whereas high expression of gE reduced the level of ISG15.

The expression profiles of free ISG15 and ISG15 conjugates were also examined in cells infected with UV-inactivated virus (UV-PRV). In UV-PRV infection, the levels of free ISG15 and ISG15 conjugates were elevated at 12 h and 24 h and correlated proportionally with MOI (Fig 1A, lanes 6–9 and 1B, lanes 6–9). Levels of free ISG15 at 36 h induced by UV-PRV were lowered than those at 24 h, probably due to the termination of signaling (Fig 1C, lanes 6–9). The lack of viral gene expression in UV-PRV infection was verified by the absence of PRV-gE protein expression (Fig 1A–1C). Collectively, these results comparing PRV and UV-PRV infection demonstrate that free ISG15 and ISG15 conjugates are initially induced by PRV infection at low MOIs, but are suppressed in a manner dependent on viral gene expression at high MOI.

### PRV gE is required for ISG15 expression

Considering the PRV-gE expression may be responsible for the upregulation of free ISG15 and ISG15 conjugates during PRV infection, we observed that the ISG15 abundance was associated with viral gene expression by confocal immunofluorescence assay (Fig 1D). We next compared the effects of PRV, UV-PRV, and gE-deleted mutant PRV infection (MOI of 1) on ISG15 transcription by RT-qPCR. All virus increased ISG15 mRNA levels 12 h after infection, however, ISG15 induction was terminated earlier for UV-PRV than PRV, and not notably terminated for gE-deleted PRV, which continued to produce high levels of ISG15 transcripts even at a late stage of infection (36 h) (Fig 1E). ISG15 transcription induced by PRV and UV-PRV infection gradually decreased might be through negative regulation of IFN-I signaling. This result indicated that PRV-gE plays a vital role in inducing ISG15 transcription during PRV infection.

We further detected the expression levels of free ISG15 and ISG15 conjugates during PRV, UV-PRV, and gE-deleted PRV infection. In consistent with the results on ISG15 transcript level, the expression of free ISG15 and ISG15 conjugates were elevated at 12 h by all viruses, whereas the level of ISG15 expression markedly decreased after 24 h in PRV and UV-PRV groups (Fig 1F). This might suggest that viral processes mediated by gE may be implicated in

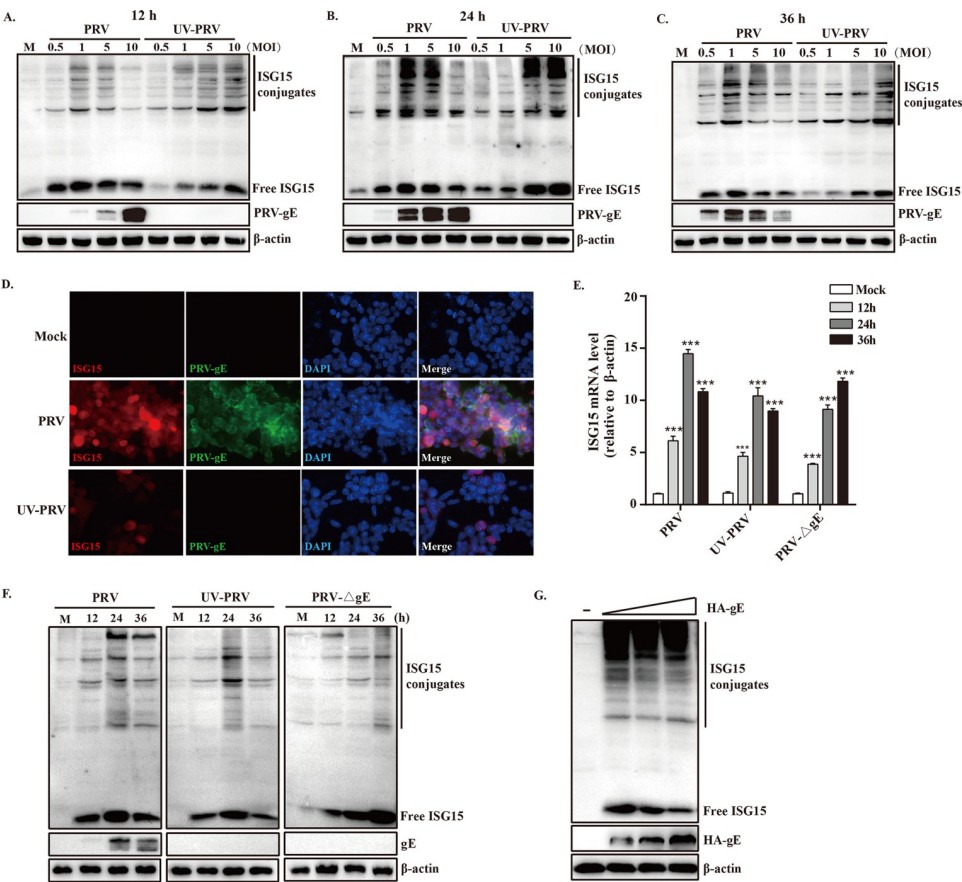

**Fig 1. Time course of ISG15 and ISGylation expression during PRV infection.** (A to C) PK15 cells were mock-infected (M) or infected with PRV or UV-PRV at different MOIs (0.5 to 10) as indicated. Cell lysates were collected at 12, 24, and 36 h after infection and immunoblotted with antibodies for ISG15, PRV-gE, or β-actin (a loading control). (D) PK15 cells were mock-infected or infected with PRV or UV-PRV at an MOI of 1 for 24 h. Cell were fixed with methanol and double-label IFA was performed with antibodies for ISG15 and PRV-gE. DAPI stain was used to stain cell nuclei. The images were obtained by confocal microscopy. (E and F) PK15 cells were mock-infected (M) or infected with PRV, UV-PRV, or gE-deleted PRV at MOI of 1 for 24 h. Total RNAs were prepared at the indicated time points and the levels of ISG15 and β-actin transcripts were determined by RT-qPCR (E). Cell lysates were also prepared and analyzed by immunoblotting (F). (G) PK15 cells were transfected with increasing amounts of plasmid expressing gE and immunoblotting was performed. ***, p < 0.01 by Student's *t* test.

the downregulation of ISG15 expression profiles. Consistent with the results shown in Fig 1B, PRV induced more ISG15 conjugates than UV-PRV at this MOI (Fig 1F, compare lanes 2 to 4 and 6 to 8). Importantly, gE-deleted virus induces a sustained increase in free ISG15, but has little effect on ISG15 conjugates, indicating that gE is required for the induction of free ISG15 expression during PRV infection. The effect of gE on free ISG15 expression was further investigated using gE-overexpressing PK15 cells. Control and gE-overexpressing PK15 cells were infected with PRV, and immunoblot results showed that overexpression of gE suppressed the induction of free ISG15, whereas had no effect on the ISG15 conjugates (Fig 1G), further supporting that gE is required for free ISG15 expression.

## ISG15 restricts PRV growth and virus release

The role of ISG15 in PRV infection was investigated by knocking out ISG15 in PK15 cells. wild-type (WT) PK15 cells and ISG15 knockout PK15 cells (ISG15$^{-/-}$) were either mock-

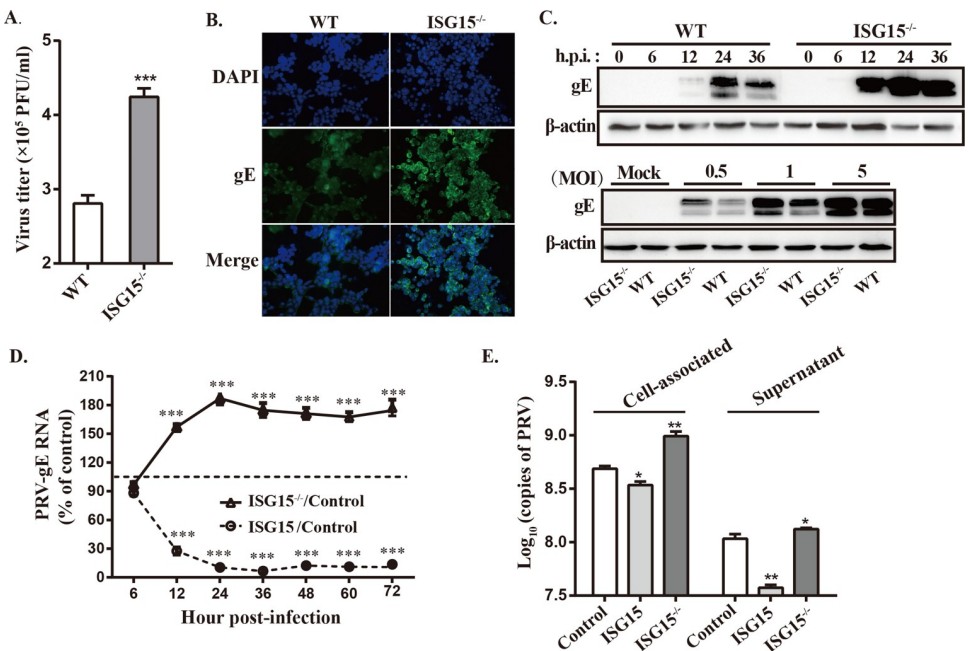

**Fig 2. ISG15 inhibits PRV replication.** (A and B) WT and ISG15$^{-/-}$ cells were infected with PRV (MOI = 1) for 24 h, and PRV titers and growth were detected by plaque assay and IFA assays. (C) PRV-gE protein levels were detected in at identified time points and different MOIs (0.5, 1, 5) by Western blot. (D) PK15 cells were transfected with ISG15-overexpressing plasmid and then infected with PRV (MOI = 1) 12 h later. PRV-gE RNA was quantified at different times post-infection in ISG15$^{-/-}$ and WT cells overexpressing ISG15 by RT-qPCR. The data represent the percentages of expression of PRV-gE in ISG15-transfected cells or ISG15 knockout cells compared with cells transfected with a control plasmid (100%). (E) The PRV titer in the culture supernatant and associated with cells was determined by plaque assay at 24 hpi. Limit of detection was 20 pfu/ml in the plaque assay. *, p < 0.05; **, p < 0.01; ***, p < 0.001 (*t*-test).

infected or infected with PRV at an MOI of 1. PRV growth were remarkably increased in ISG15$^{-/-}$ cells, as evidenced by viral plague, IFA and Western blot assays (Fig 2A–2C), suggesting that ISG15 inhibited PRV replication. These results were also confirmed in mouse embryo fibroblasts (MEF) cells (S1 Fig).

Additionally, it has been reported that ISG15 may affect virus entry [35] or release [41,42]. Thus, two experiments were carried out to gain information about the role of ISG15 in those steps of PRV replication. First, PK15 cells were transfected with a plasmid expressing ISG15 or a control plasmid before being infected with PRV, meanwhile, ISG15$^{-/-}$ cells were also infected with PRV. The RNA level of PRV-gE, a protein encoded by late gene [43], was quantified at various times post-infection. We found that a significant RNA reduction in the ISG15-overexpressing cells compared to control cells starting 6 hpi, while loss of ISG15 significantly enhanced PRV-gE RNA level (Fig 2D). This result showed that ISG15 restricts PRV growth at a post-entry stage of infection. Furthermore, the virus associated with cells and released to the supernatant were also quantified in control- or ISG15-transfected WT cells and ISG15$^{-/-}$ cells. As shown in Fig 2E, a significant decrease was observed in virus titers from cell-associated fraction of ISG15-transfected cells compared to control-transfected cells, while an obvious increase in ISG15 silencing cells. Moreover, more than 3-fold reduction was observed in virus titer from the supernatant fraction of ISG15-transfected cells compared to the same fraction of cell-associated cells (Fig 2E). These results indicate that ISG15 limits PRV replication occurring before virus release.

## Antiviral activity of ISG15 against PRV is due to free ISG15

To further determine whether ISG15 achieves its antiviral effect against PRV via free or conjugated form, two different experiments were carried out. First, due to the fact that exposure of the C-terminal Gly-Gly motif is essential for conjugation of ISG15 to substrates [21], these two residues were replaced with Ala using site-directed mutagenesis to generate a mutate plasmid ISG15AA employed as a control. PK15 cells were transfected with an empty vector, a ISG15-expressing plasmid, or an ISG15AA expressing plasmid. After 24 h post-transfection (hpt), the cells were infected with PRV and then viral protein expression and viral titer were measured. As expected, we found a significant decrease in PRV-gE expression level, as well as PRV titer (3.2-fold) (Fig 3A), compared to cells transfected with empty vector. Notably, there wasn't any significant differences detected in PRV titer and protein levels between these cells transfected with ISG15 and ISG15AA plasmid (Fig 3A), suggesting ISGylation did not seem to play an antiviral effect against PRV. Similar results were obtained from a parallel transfection / infection in ISG15$^{-/-}$ cells (Fig 3B). These data suggest that ISG15 exerts its antiviral activity against PRV through free ISG15-dependent or ISG15 conjugation-independent mechanisms.

The effect of free ISG15 on PRV growth was further investigated by depleting UbE1L, a known E1 enzyme for ISGylation [44]. Prior to infection with PRV, PK15 cells were transfected with either a control siRNA or a siRNA targeting UbE1L, and expression of free ISG15 and conjugates were measured by immunoblotting. We observed that UbE1L-knockdown

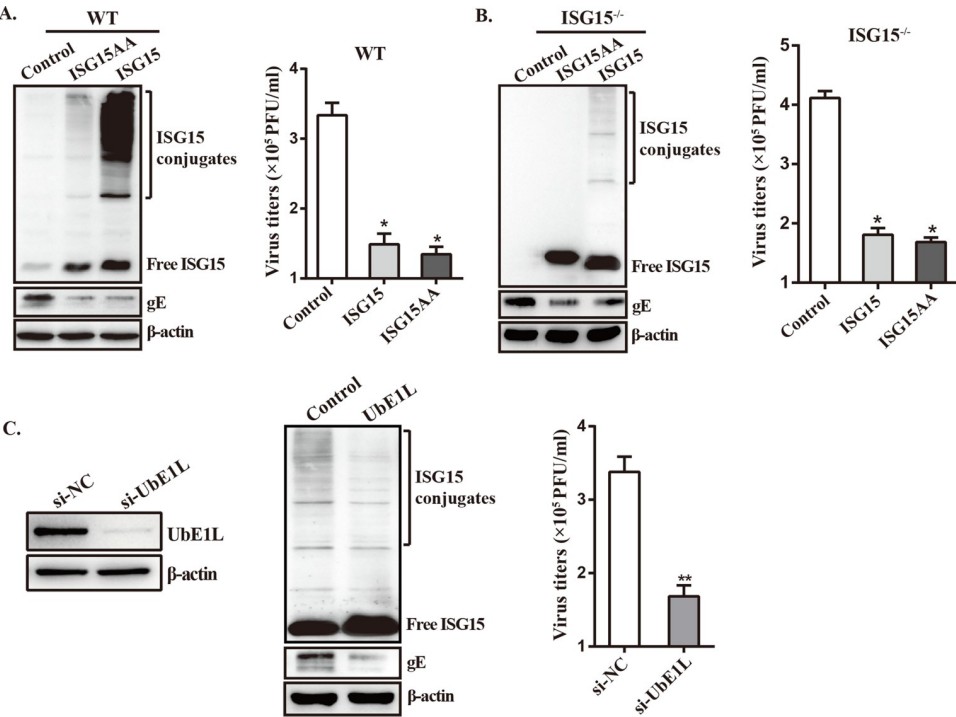

**Fig 3. Antiviral activity of ISG15 against PRV relies on free ISG15.** (A) PK15 cells were transfected with either an empty control plasmid, an ISG15-expressing plasmid, or a plasmid expressing ISG15 mutant (ISG15AA). After 24 hours, the cells were infected with PRV (MOI = 1). Total proteins were collected 24 hpi, and Western blot was used to determine the expression of free ISG15 and ISGylation protein. The PRV titer was detected by plaque assay. (B) As in panel A, ISG15$^{-/-}$ cells were transfected and infected, and the ISG15 protein expression and virus titer were determined. (C) PK15 cells were transfected with either control siRNA or UbE1L siRNAs and infected with PRV (MOI = 1) for 24 h. The knockdown efficiency of UbE1L was determined by Western blot. *, p < 0.05; **, p < 0.01 (*t*-test).

cells displayed higher levels of free ISG15 than control cells, but a slight decrease in the expression levels of ISG15 conjugates (Fig 3C). This might be due to the fact that ISGylation was incompletely inhibited by UbE1L knockdown. However, a significant drop in viral titer was found in UbE1L-knockdown cells compared with the control cells (Fig 3C), suggesting free ISG15 exerted anti-PRV activity. Meanwhile, the antiviral activity of free ISG15 was also confirmed in MEF cells (S2 Fig).

Collectively, these results indicate that ISG15 inhibits PRV replication in a free ISG15-dependent manner, rather than ISGylation.

## ISG15 knockout decreases PRV-induced IFN-β production

Like other alphaherpesviruses, PRV establish persistent infections rely partly on their ability to inhibit IFN-I signaling [13]. To determine if free ISG15 is a factor involved in the innate antiviral response, we firstly test the effect of free ISG15 on type I IFN production. We complemented the ISG15$^{-/-}$ cells with conjugation-deficient ISG15 mutant ISG15AA as a control, and no significant difference was observed for ISG15 expression between ISG15$^{-/-}$ cells transfected with ISG15AA-expressing plasmid and WT cells, indicating successful rescue of ISG15 (Fig 4A). The cells responded to PRV infection with induction of IRF3(S396) phosphorylation, but

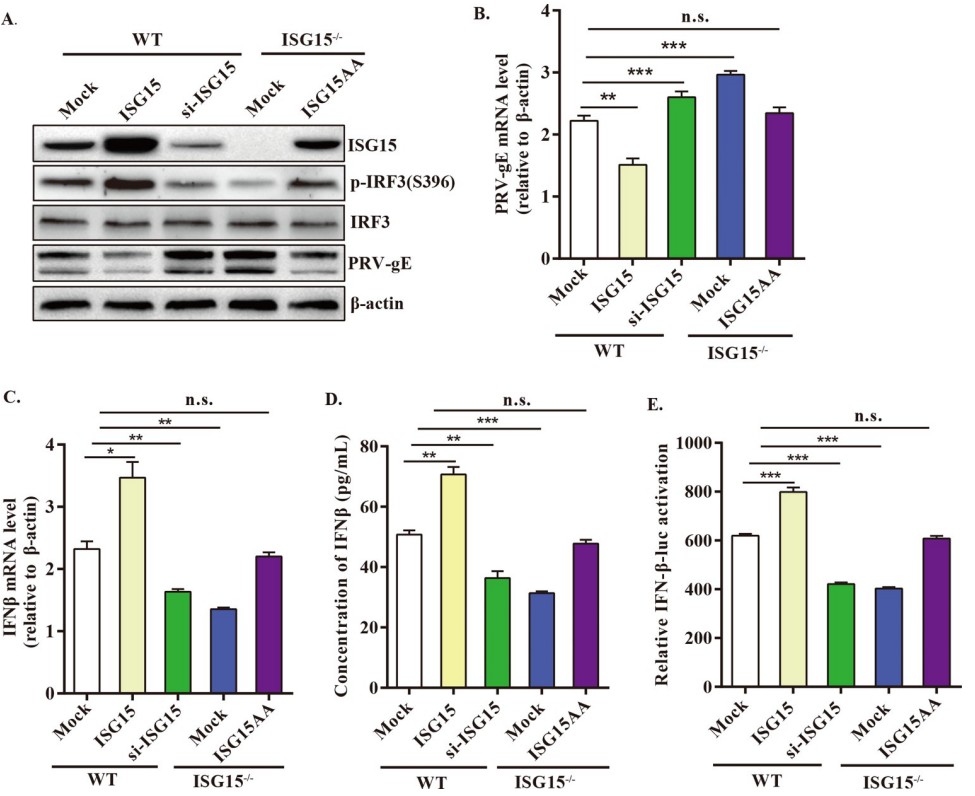

**Fig 4. ISG15 deficiency decreases IFNβ production.** (A to C) WT cells were transfected with empty vector, plasmid overexpressing ISG15, or siRNA, and ISG15$^{-/-}$ cells were transfected with plasmid expressing ISG15AA or empty vector. At 24 hpt, cells were infected with PRV (MOI = 1) for 24 h. The cell lysates and total RNA were harvested respectively. (A) The protein levels of IRF3, pIRF3 (S396), PRV-gE and ISG15 were analyzed by Western blot. (B)The mRNA levels of PRV-gE and IFN-β were detected by RT-qPCR. (D) WT and ISG15$^{-/-}$ cells were transfected with expression plasmid ISG15, siISG15, ISG15AA or empty vector alongside IFN-β luciferase reporter gene (IFN-β-Luc). After 12 hpt, cells were infected with PRV for 24 h, and the promoter activity of IFN-β was performed by dual-luciferase reporter (DLR) gene assay. (E) As in panel A, the cellular supernatant was collected to analyze the protein concentration of IFN-β by ELISA. n.s., no significant difference; *, $p < 0.05$; **, $p < 0.01$; ***, $p < 0.001$ (t-test).

this event was suppressed upon ISG15 knockdown or knockout at 24 hpi (Fig 4A). Reduced levels of IRF3 activation are due to enhanced viral transcription/translation, resulting in enhanced viral growth upon ISG15 depletion (Fig 4A and 4B). Elevated levels of IRF3 activation corresponded with significantly enhanced induction of IFN-β transcripts, which induction of IFN-β mRNA was also increased (Fig 4C). Meanwhile, ISG15 depletion statistically diminished PRV-induced IFNβ production and release (Fig 4D). Moreover, we found that IFN-β promoter activity was dramatically enhanced in ISG15-expressing WT cells, while reduced in siISG15-expressing WT cells, compared with samples transfected with empty vector (Fig 4C), further suggesting that free ISG15 accelerates IFN-β production. Overall, these findings indicate that ISG15 knockout attenuates PRV-triggered IFN-β production and release, subsequently potentiating PRV replication.

## ISG15 deficiency impairs the IFN𝛼-mediated antiviral activity

Previous reports have indicated that IFNα treatment of ISG15-deficient patient cells exhibited increased resistance to several viral infection [22]. Considering that ISG15 promotes IFN-β production during PRV infection, we wonder whether ISG15 is involved in type I IFN-mediated antiviral effect against PRV. We firstly compared the ISG15 expression patterns induced by PRV infection or IFNα stimulation, and found that PRV-induced ISGylation differs to some extent from that of IFNα, as well as some specific bands were apparent in PRV-infected cells (Fig 5A). This might be involved in some strategies used by PRV to antagonize IFNα-mediated antiviral effect. Subsequently, we found a significant increase in viral RNA and viral titers in ISG15$^{-/-}$ cells with or without IFNα stimulation compared with WT controls (Fig 5B and 5C), implying that complete loss of ISG15 impairs IFNα-mediated antiviral activity against PRV.

With the aim of confirming that the effect observed was specifically dependent on free ISG15, we complemented the ISG15$^{-/-}$ cells with conjugation-deficient ISG15 mutant ISG15AA as a control. No significant difference was observed for PRV-gE expression between ISG15$^{-/-}$ cells transfected with ISG15AA-expressing plasmid and WT cells, indicating successful rescue of ISG15 (Fig 5D). Results showed that, as expected, transfected with ISG15AA led to a high suppression of the PRV-gE expression (Fig 5D), suggesting ISG15-deficient cells with or without IFNα stimulation are more susceptible to PRV. Notably, substantial suppression of IFNα-induced ISRE promoter activity was also observed in PRV-infected ISG15$^{-/-}$ cells by a dual-luciferase reporter assay (Fig 5E), implying that ISG15 deficiency can interfere with IFN signaling. To further confirm the inhibition effect of ISG15 on ISRE transcription, mRNA expression levels of three common ISGs were detected by RT-qPCR, including *interferon induced protein with tetratricopeptide repeats* 1 (IFIT1), *oligoadenylate synthetase* 1 (OAS1), and *myxovirus-resistance* A (MxA). Results showed that mRNA levels of IFIT1, OAS1 and MxA induced by PRV were significantly reduced in ISG15$^{-/-}$ cells (Fig 5F). Altogether, these data support that ISG15 deficiency interferes with type I IFN signaling and impairs IFNα-mediated antiviral effect against in PRV.

## ISG15 deficiency suppresses phosphorylation and dimerization of STAT1 and STAT2

Phosphorylation of STAT1 and STAT2 is required for activation of IFN-I-mediated antiviral response [45], and STAT degradation is a common mechanism of viral IFN antagonism [46]. We next sought to investigate the mechanism by which ISG15 interferes with IFN-I signaling. Since ISG15 deficiency inhibits IFN-I signaling pathway, the ability of ISG15 to affect STAT1 and STAT2 phosphorylation after PRV infection was initially investigated. Western blot

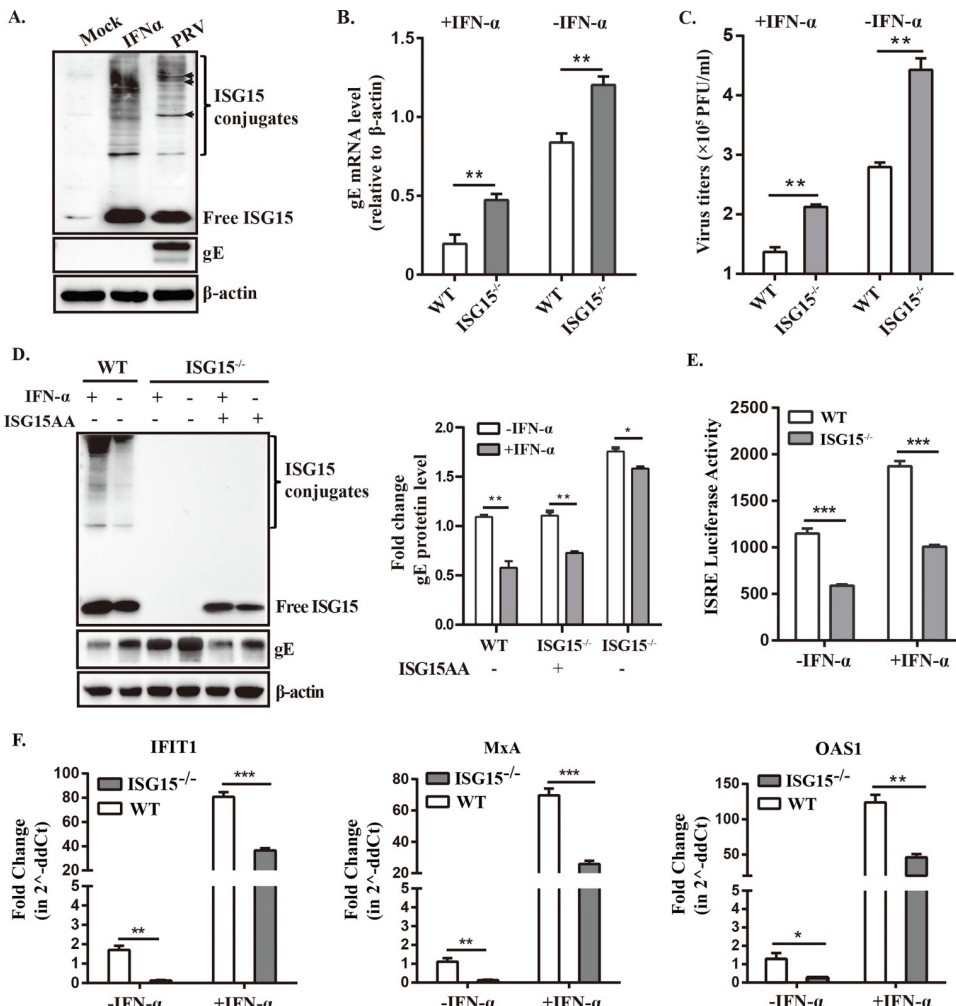

**Fig 5. ISG15 deficiency impairs IFNα-mediated antiviral activity against PRV.** (A) Total protein from mock infected, stimulated with IFNα (1000 U/ml) or infected with PRV (MOI = 1) were collected at 12 hpi, fractionated by SDS-PAGE, and analyzed by immunoblotting using antibodies specific for total ISG15, gE, or β-actin. (B and C) Prior to infection with PRV (MOI = 1), WT and ISG15$^{-/-}$ cells were either treated with IFNα (1000 U/ml) or left untreated for 12 h. Total RNA and supernatant were harvested 24 hpi to analyze gE mRNA level (B) and viral titers (C) by RT-qPCR and plaque assays. (D) ISG15$^{-/-}$ cells were transfected with either a plasmid expressing ISG15 mutant or an empty plasmid. After 12 hpt, WT and ISG15$^{-/-}$ cells were treated with IFNα (1000 U/ml) or left untreated before infecting with PRV. The expression of total ISG15 and gE were determined by Western blot. (E) WT and ISG15$^{-/-}$ cells were co-transfected with the ISRE reporter plasmid and the Renilla luciferase control plasmid (pRL-TK). Twelve hours post-transfection, cells were infected with PRV at an MOI of 1 for 24 h and pre-treated with IFNα (1,000 U/ml) or left untreated for 12 h. Relative luciferase activity was measured by dual-luciferase reporter (DLR) assay. (F) The mRNA levels of ISGs, including IFIT1, MxA and OAS1 were measured by RT-qPCR. Fold change in mRNA levels relative to the untreated group was calculated using the $2^{\triangle\triangle CT}$ method, and the $\beta$-actin gene was used as the housekeeping gene. Limit of detection was 20 pfu/mL in the plaque assay. The data are presented as means standard error of the mean ± (SEM) of at least three independent experiments. $^{*}$, p < 0.05; $^{**}$, p < 0.01; $^{***}$, p < 0.001 (t-test).

analysis showed that phosphorylation of endogenous STAT1 and STAT2 proteins were degraded in PRV-infected ISG15$^{-/-}$ cells, and the protein levels of total STAT1 and STAT2 remained constant before and after IFNα treatment (Fig 6A). To confirm the effect observed was specifically dependent on ISG15 deficiency, we complemented the ISG15$^{-/-}$ cells with ISG15AA transfection, showing no difference compared with WT cells (Fig 6A). This result clearly showed that ISG15 deficiency suppressed the activation of STAT1 and STAT2.

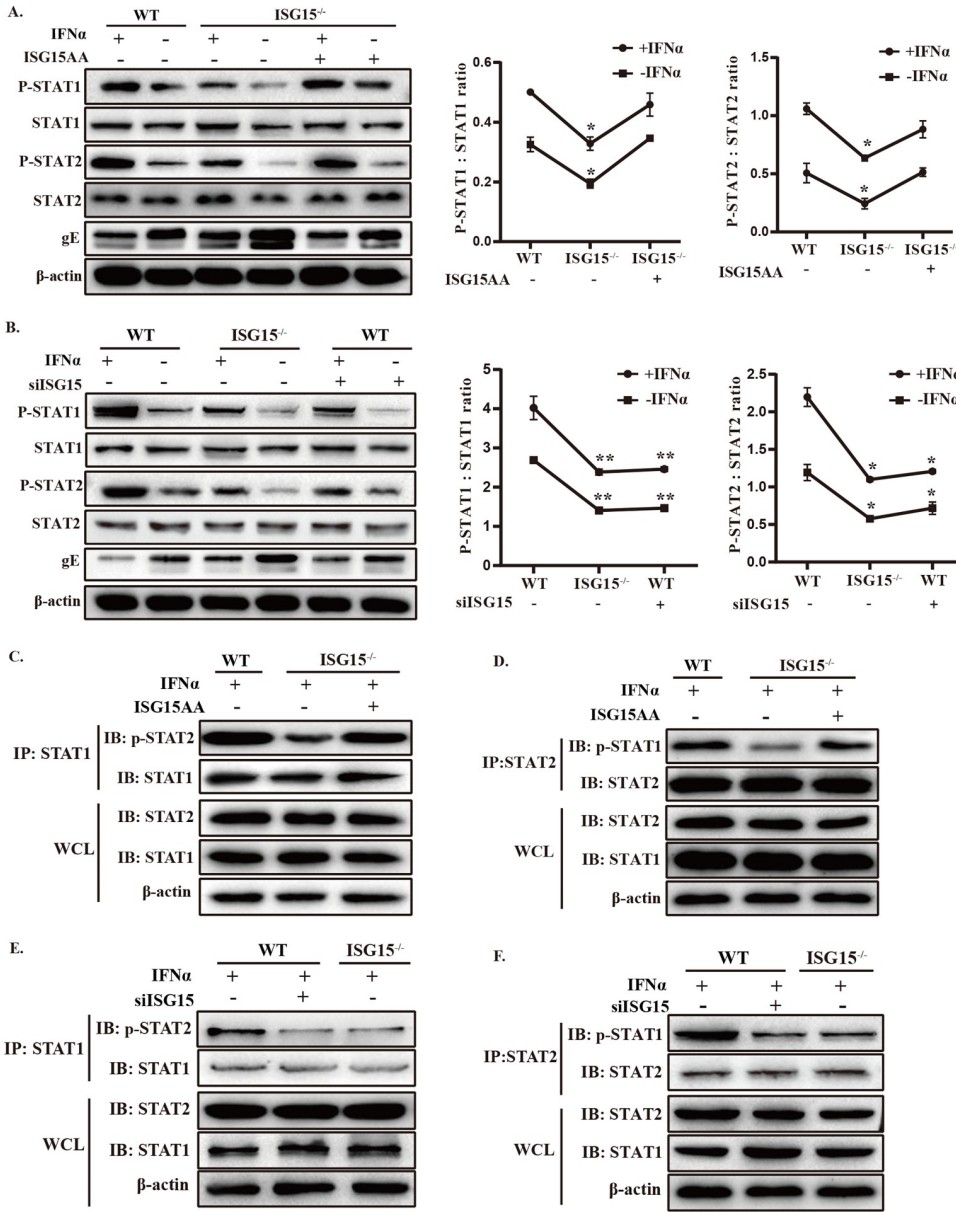

**Fig 6. Knockout of ISG15 prevents the activation of STAT1 and STAT2.** (A and B) WT and ISG15[-/-] cells were transfected with ISG15AA or siISG15 for 24h, and the cells were treated or un-treated with IFNα (1000 U /ml) and then infected 12 h later with PRV (MOI = 1). At 24 hpi, the relative pSTAT1 and pSTAT2 levels were analyzed by Western blotting with the specified antibodies. The pSTAT1/STAT1 and pSTAT2/STAT2 protein levels were calculated by ImageJ software. (C to F) IP detection of STAT1 and STAT2 heterodimer in WT and ISG15[-/-] cells after PRV infection. The cells were transfected with ISG15AA in ISG15[-/-] cells or siISG15 in WT cells before IFNα treatment, and then infected with PRV (MOI = 1) for 24 h. IP with STAT1 or STAT2 antibody and then blotting with antibody against STAT2 (Tyr 690) or STAT1 (Tyr 701). *, $p < 0.05$; **, $p < 0.01$ (*t*-test).

Furthermore, opposite results were observed in siISG15-transfected WT cells (Fig 6B). Consistent with the results above, no significant difference was observed in the ratio of p-STAT1/STAT1 and p-STAT2/STAT2 between ISG15[-/-] cells and siISG15-transfected WT cells (Fig 6B). These data indicated ISG15 deficiency blocked endogenous STAT1 and STAT2 phosphorylation independent of IFNα treatment.

Once activation is triggered by type I IFN, STAT1 and STAT2 form a heterodimer and associate with IRF9 to form the mature ISGF3 complex [47]. To examine the effect of free ISG15 on the formation of STAT1 and STAT2 heterodimerization, we performed Co-IP assays to detect the interaction between ISG15 and STAT1/STAT2. IP with STAT1 antibody followed by immunoblotting with STAT2-Y690 antibody showed ISG15 deletion inhibited the STAT1--STAT2 interaction (Fig 6C), and the same result was observed in samples with STAT2 antibody and then blotting with STAT1-Y701 antibody (Fig 6D). Moreover, no significant difference was observed between WT cells and ISG15AA-transfected ISG15$^{-/-}$ cells, indicating the inhibition role of free ISG15 on the formation of STAT1 and STAT2 heterodimer (Fig 6C and 6D). Similar results were obtained in siISG15-transfected WT cells, further confirming ISG15 knockout inhibit the formation of STAT1 and STAT2 heterodimerization (Fig 6E and 6F). The above data suggested that ISG15 facilitates phosphorylation and dimerization of STAT1 and STAT2.

### ISG15 deficiency blocks IFN-mediated nuclear translocation of STAT1/STAT2 by preventing ISG15-STAT2 interactions

Next, we set out to determine how ISG15 affects the activation of STAT1 and STAT2. We first examine the direct interactions between ISG15 with STAT1/STAT2 by Co-IP assays. Immunoprecipitation results showed that ISG15 was only able to interact with STAT2 (Fig 7A and 7B), hinting that STAT2 is the crucial component connecting STAT1/STAT2 to ISG15. The resulting phosphorylation of STAT1 and STAT2 allows their heterodimerization and associated with IRF9, forming the ISGF3 (STAT1/STAT2/IRF9) complex that subsequently translocated to the nucleus to initiate ISRE-dependent transcription. To further clarify the positive regulation of ISG15 on IFN-I signaling, the formation of ISGF3 heterotrimers was also examined by Co-IP. Results showed ISG15 deficiency impaired the formation of ISGF3 (Fig 7C).

Next, we examined whether the impaired STAT1/STAT2 phosphorylation correlated with the inhibition of STAT1/STAT2 nuclear translocation or a reduction in steady state levels of ISGF3 members. Subcellular fractionation and Western blot analysis showed that ISG15 deficiency prevents STAT1 and STAT2 nuclear translocation in the absence of IFNα treatment (Fig 7D), indicating that ISG15 deficiency blocks the nuclear translocation of STAT1/STAT2. We further determined whether ISG15 deficiency is involved in the ISRE activity by a dual-luciferase reporter gene assay, and identified that ISG15$^{-/-}$ cells significantly reduced luciferase activity driven by the ISRE (Fig 7E), confirming ISG15 deficiency prevents the ISGF3-induced ISRE activity. These findings indicated that ISG15 knockout inhibited the nuclear translocation and ISGF3-induced ISRE activity by blocking the ISG15-STAT2 interaction. Taken together, these results suggested that ISG15 interacts with STAT2, which is important for ISG15 to enhance the activation of ISRE.

### ISG15 knockout mice are highly sensitive to PRV infection

PRV can infect a variety wide of mammals including rodents, thus mouse model has been widely used to study PRV pathogenesis [48]. To investigate whether ISG15 possesses an antiviral role *in vivo*, we infected ISG15 knockout (ISG15$^{-/-}$) mice with PRV for 7 days, and analyzed the clinical symptoms, survival rate, viral loads and pathological changes. The results showed that the ISG15$^{-/-}$ mice displayed severe typical neurological symptoms, including greatly reduced activity and pruritus at 3 days post infection (dpi), began to die at 4 dpi and all die on day 6 (Fig 8A). Whereas the PRV-infected WT mice developed only mild symptoms at 4dpi under the same condition, and started to die from the day 5 and over 50% of them eventually survived from the challenge (Fig 8A). Additionally, the ISG15 expression was remarkably

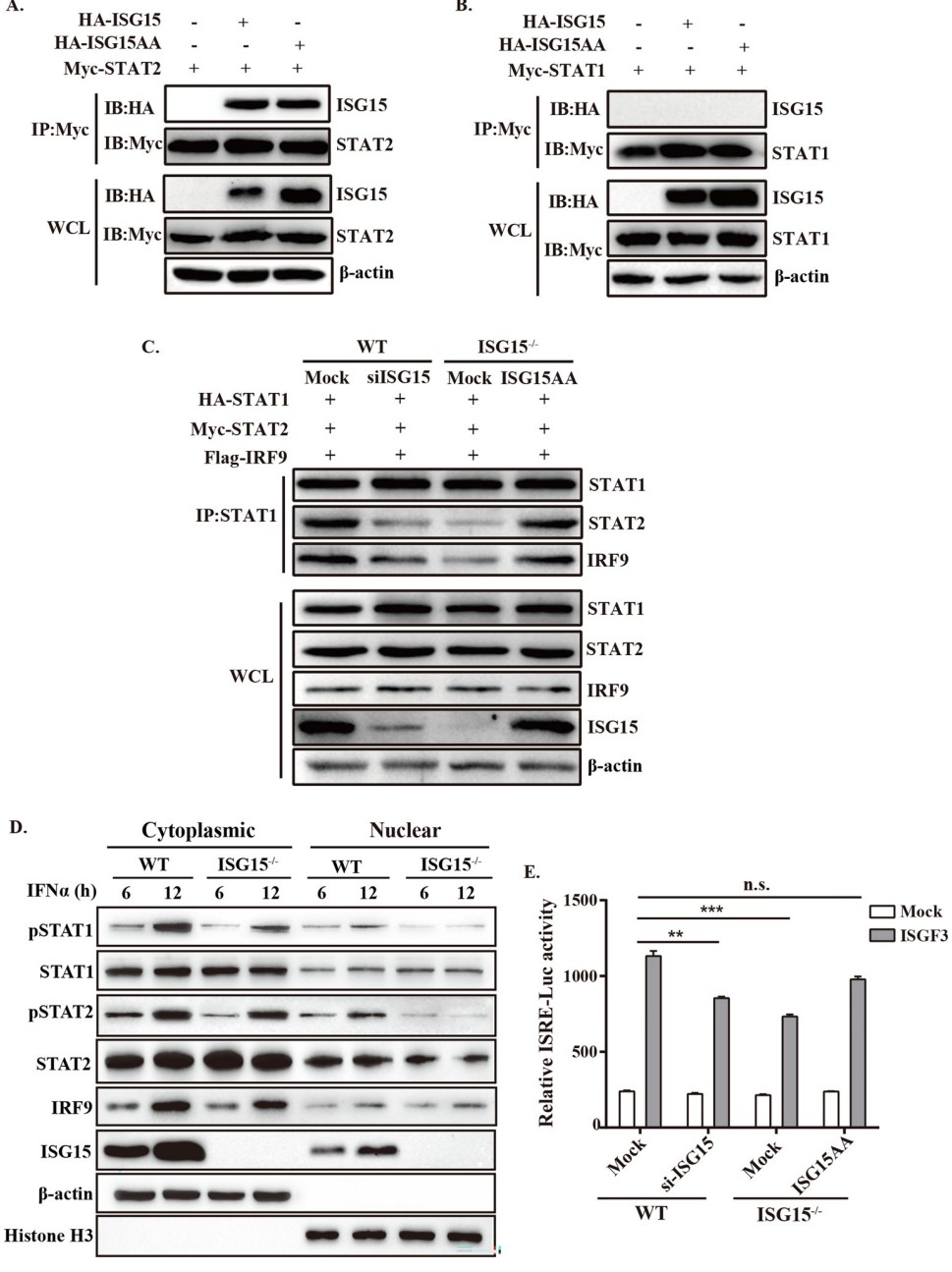

**Fig 7. ISG15 facilitates activation of STAT1/STAT2 mainly through STAT2.** (A and B) IP analysis of the interaction between STAT1/STAT2 and ISG15. IB analysis of immunoprecipitations of ISG15$^{-/-}$ cells co-transfected with HA-ISG15AA, HA-ISG15 or Myc-STAT2/STAT1 expression plasmids. (C) WT and ISG15$^{-/-}$ cells were co-transfected with HA-STAT1, Myc-STAT2, Flag-IRF9 and si-ISG15 (in WT) or ISG15AA (in ISG15$^{-/-}$), and the cells were infected 12 h later with PRV (MOI = 1). At 24 hpi, IP with STAT1 antibody and then blotting with antibody against STAT2 (Tyr 690), STAT1 (Tyr 701) and IRF9. (D) Subcellular fractionation and Western blotting were used to identify the distribution and expression level of STAT1 and STAT2 in WT and ISG15$^{-/-}$ cells. The same blot was incubated with antibodies against $\beta$-actin and histone H3 as controls for loading and fractionation. (E) WT and ISG15$^{-/-}$ cells were co-transfected with si-ISG15 or ISG15AA, along with firefly luciferase reporter plasmid and pRL-TK. After 12 hpt, the cells were infected with PRV (MOI = 1) for 24 h. Relative luciferase activity was measured by dual-luciferase reporter (DLR) assay. n.s., no significant difference, **, p < 0.01, ***, p < 0.001 (*t*-test).

upregulated during PRV infection in WT mice (Fig 8B), consistent with the results from our cell model.

Because PRV infection mainly causes neurological and respiratory symptoms, and encephalitis is a key factor contributing to animal death [49]. As expected, ISG15[-/-] mice displayed more susceptible to PRV than WT mice, as evidenced by increased viral loads within the brains and lungs of infected ISG15[-/-] mice (Fig 8C). Next, to detect the degree of encephalitis and

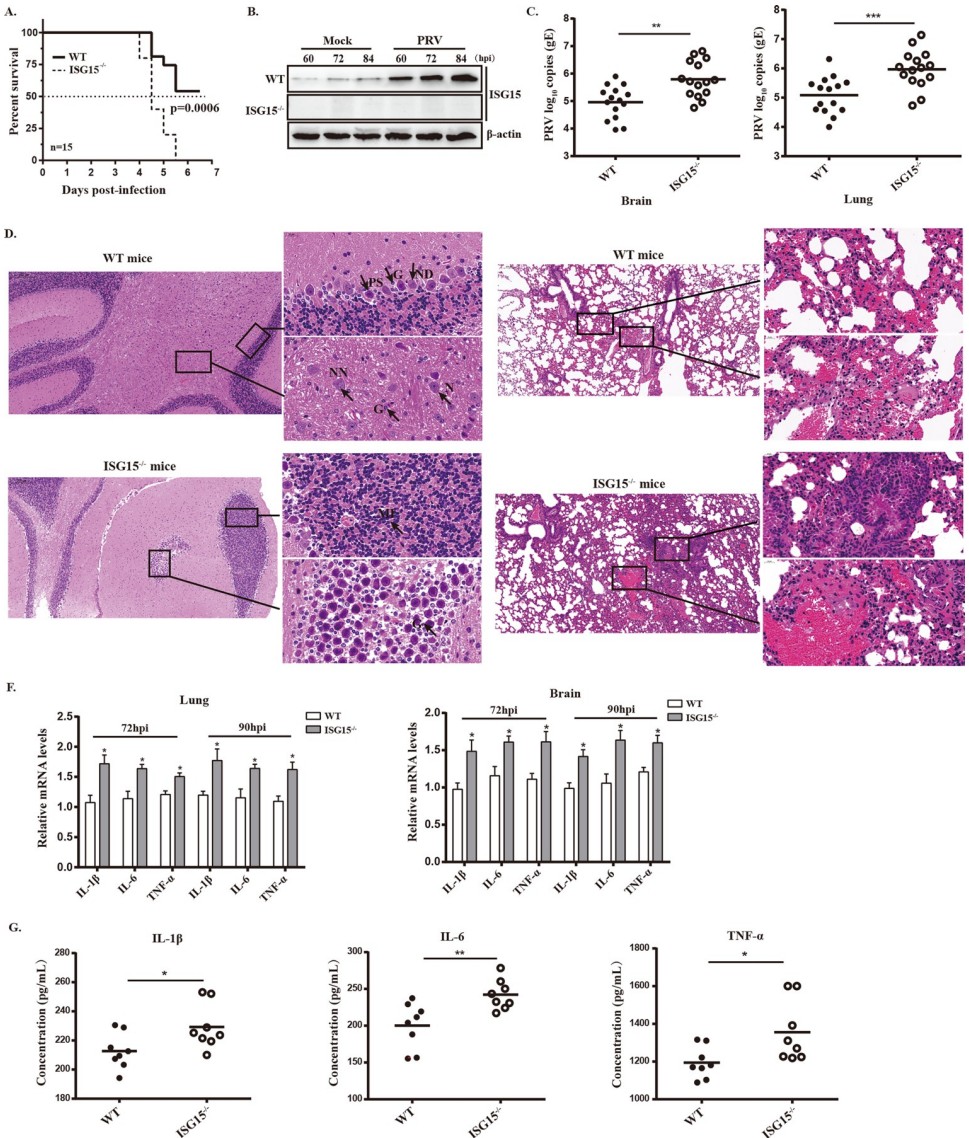

**Fig 8. PRV challenge assay in vivo.** Seven-week-old male ISG15[-/-] mice (n = 15) and WT mice (n = 15) were inoculated with $5\times10^3$ TCID$_{50}$ of PRV-QXX subcutaneous. (A) Survival of the infected WT and ISG15[-/-] mice was monitored until day 6 after infection. Statistical significance was determined by the log-rank test. (B) Expression of ISG15 and gE in the brain tissue from PRV-infected mice. (C and D) Viral copies of brain and lung tissues were detected by RT-qPCR. (E) The histopathological features of brains and lungs of the infected WT and ISG15[-/-] mice. The tissues were sectioned and stained with hematoxylin-eosin. Magnified images of the regions with black rectangles in infected WT and ISG15[-/-] mice, respectively. Representative images are shown: ND: nuclear disintegration of Purkinje cell; N: neurons; NN: necrotic neurons; G: glial cells; PS: shrinkage of Purkinje cells; P: Purkinje cells; PN: necrotic Purkinje cells; MI: mononuclear cellular infiltration. (F and G) The mRNA and protein expressions of IL-6, TNF-α and IL-1β in different tissues and serum of infected mice were determined by RT-qPCR and ELISA, respectively.

pneumonia in the PRV-infected mice, the histopathological analysis of the brains and lungs at 4dpi were performed. As illustrated in Fig 8D and 8E, the brains of ISG15$^{-/-}$ mice show greater inflammatory damage, necrotic neurons, more glial cells and more obvious microgliosis and hyperemia compared with WT mice (Fig 8D). Histological analyses of infected lungs showed greater inflammatory cells infiltration, severe congestion and higher level of lung tissue impairment in ISG15$^{-/-}$ mice in comparison with WT mice (Fig 8E). These results indicated that ISG15 deletion aggravated virus-induced pathogenicity during PRV infection *in vivo*.

Cytokines are crucial in combating viral infection and are involved in the regulation of immune and inflammatory responses, including interleukin 1β (IL-1β), interleukin-6 (IL-6) and tumor necrosis factor alpha (TNF-α) [50]. Thus, we also evaluated whether ISG15 deficiency influences cytokines production in the brains and lungs of PRV-infected mice. The mRNA levels of IL-1β, IL-6 and TNF-α were significantly higher in the brains and lungs of ISG15$^{-/-}$ mice as compared with those of WT mice at different time points post infection (Fig 8F). Moreover, the protein concentrations of IL-1β, IL-6 and TNF-α from serum of the infected mice were measured using ELISA assay, and we found ISG15-deficient mice displayed higher cytokines concentrations (Fig 8G). This suggested that ISG15$^{-/-}$ mice produced more inflammatory cytokines, which is responsible for severe brain and lung pathology. These data demonstrate that ISG15 exhibits antiviral function *in vivo*.

## Discussion

The function of ISG15 in viral immunity remains an area of active investigation [25,39,51,52]. To date, the PRV-host interactions that induces ISG15 upregulation and the impact of the free ISG15 and ISG15 conjugates on PRV replication remained unexplored. Here, we established that although the ISG15 abundance were triggered by PRV infection, they are subsequently abrogated by viral gene expression at high MOI. A possible reason for this finding is that viral gene expression was high enough to prevent ISG15 induction. Our analysis with gE-deleted mutant virus demonstrated that PRV-gE played a central role in inducing free ISG15. Notably, although gE effectively reduced free ISG expression, the level of free ISG15 and ISGylated proteins during PRV infection largely depended on the MOI.

Although the antiviral role of ISGylation has been reported in several viruses, studies on free ISG15 interferes with the antiviral functions are limited to a few examples. Our data provide evidence supporting the antiviral roles of free ISG15 during PRV infection using a ISG15$^{-/-}$ cell line. Additionally, we noticed that ISG15 had to accumulate in large amounts before virus replication to carry out its anti-PRV role. To discern the effects of free ISG15 and ISG15 conjugates on PRV replication, WT and ISG15$^{-/-}$ cells was overexpressed with a conjugation-deficient ISG15 mutant ISG15AA, suggesting that free ISG15 significantly inhibited PRV growth. Another powerful evidence is that UbE1L, a specific ISGylation enzyme, was depleted by shRNA in WT cells. These data consistently showed an inverse relationship between the expression of free ISG15 and PRV growth, further indicating that free ISG15 inhibits PRV growth. It is possible that when ISG15 is expressed at high levels before virus replication, no viral proteins are present to counteract the ISG15 antiviral activity (Fig 1). Another possibility is that ISG15 accumulation may promote IFN-I signaling and/or the expressions of ISGs to combat viral infection. It should be noted that the negative correlation between PRV infection and ISG15 expression was also observed from the infected mice (Fig 8B and 8C), pointing to the antiviral role of ISG15 in PRV infection *in vivo*.

We further demonstrated here that complete loss of ISG15 resulted in a reduced IFN-β production by inhibiting IRF3 activation, and impaired IFNα-mediated antiviral response against PRV. Type I IFN signaling is a critical for controlling viral infection. Recent investigations

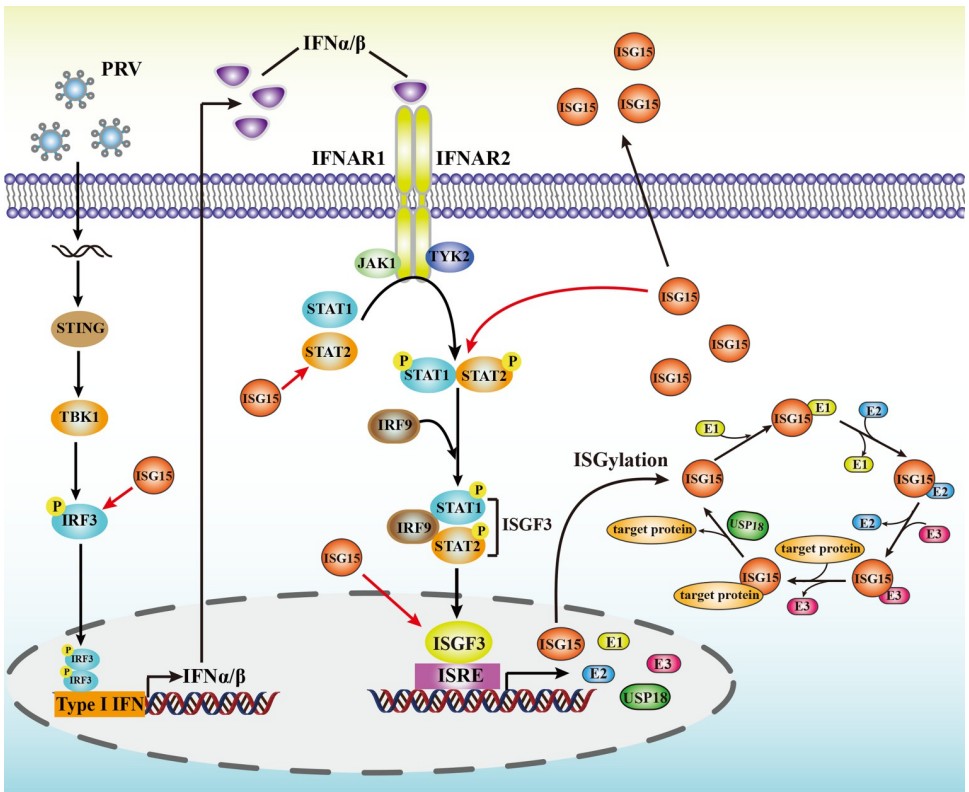

**Fig 9. A mechanism model showing ISG15 facilitates type I IFN-mediated antiviral response.** Our work demonstrates that ISG15 plays a positive feedback role in type I IFN-mediated antiviral activity against PRV.

demonstrated that ISG15 acted as a negative regulator of type I IFN signaling exerted antiviral response during viral infection [22,39]. However, our results provided some evidence supporting ISG15 as a positive regulator of IFNα-mediated antiviral response against PRV as following: 1) ISG15 deficiency inhibits IFN-β production and promotes PRV replication; 2) ISG15 deficiency impairs the IFNα-mediated antiviral activity against PRV; 3) ISG15 deficiency prevents the phosphorylation of STAT1/STAT2 and nuclear translocation of STAT1/STAT2 heterodimers; and 4) ISG15 deficiency blocked the ISGF3 formation and several ISGs transcription. Collectively, these data showed that ISG15 deletion reduces the production of the type I IFN and subsequently prevents the activation of STAT1 and STAT2 in response to PRV infection. These findings demonstrate ISG15 as a key positive regulator in IFN-I signaling and confirm its importance in the host defense against PRV infection (Fig 9), which may be more broadly against other viruses as well.

We found that ISG15 was involved in two crucial steps in IFN-I signaling, including the activation of STAT1 and STAT2 and the ISGF3 formation (Figs 6 and 7). This may be partly because ISG15 knockout inhibited the IFN-I signaling pathway by blocking the activation of STAT1 and STAT2 (Fig 6A and 6B). Since ISG15 mainly localizes in the cytoplasm (Fig 7D), we found the mechanism for ISG15 to impact this step is through interactions between ISG15 and STAT2. The result that the lack of ISG15 decreases STAT1 and STAT2 phosphorylation with hindering formation of STAT1 and STAT2 heterodimer suggests that ISG15 may be involved in the formation of the ISGF3 heterotrimer. Although the regulation of the ISGF3-mediated transcription of ISGs in the nucleus in not well understood, we demonstrate that ISG15 carries out a critical positive regulator in this process. ISG15 seems to function by

enhancing the ISGF3 recruitment to the promoter of ISGs and promoting the transcription of ISGs, this is complex. ISGF3 complex is essential for ISRE activation. Thus, we described that ISG15 may act as a regulator promoting ISGF3 to its ISGs promoters for efficient gene transcription (Fig 9). A similar mode of action is also observed in Bclaf1 that regulated the type I IFN responses and was degraded by alphaherpesvirus US3 [53].

From the above results, we concluded that ISG15 contributes to the IFNα-mediated antiviral response. Supporting this conclusion, ISG15$^{-/-}$ mice are highly susceptible to PRV infection than WT mice, as evidenced by increased mortality rates and viral loads. Moreover, ISG15 knockout mice displayed more severe encephalitis and excessive production of cytokines during PRV infection. The increased susceptibility to PRV induced lethality seemed to correlate with a cytokine storm exhibiting severe encephalitis and pneumonia. Our results contrast with the previous studies that ISG15-deficient patients who display no enhanced susceptibility to viruses *in vivo* [54]. This reflects the ISG15 function may vary depending on the virus and host species.

Overall, our findings confirm the principal role for ISG15 as a positive regulator of type I IFN signaling by facilitating STAT1 and STAT2 activation and nuclear translocation, which provide a viable option for developing therapeutic target for controlling PRV.

## Materials and methods

### Ethics statement

C57BL/6N (WT) and ISG15$^{-/-}$ mice were purchased from Cyagen Biosciences, Inc. (Guangzhou, China). Animal experiments were performed in accordance with protocols approved by the National Research Center for Veterinary Medicine (Permit 20180521047).

**Cell culture and virus.** Porcine kidney epithelial cells (PK15) and mouse embryo fibroblasts cells (MEF) were cultured at 37˚C in 5% $CO_2$ in Dulbecco's modified Eagle medium (DMEM; Gibco) supplemented with 10% fetal bovine serum (FBS; Gibco) and 1% penicillinstreptomycin (DingGuo, Beijing, China). The PRV-QXX virus and gE-deleted PRV strains were preserved in our laboratory. For experiments, PRV was amplified in PK15 cells, and virus titers were determined using a plaque assay, as previously described [40]. The infectivity of each sample was assayed by plaque titration.

**Quantitative RT-PCR and Western blot.** According to the protocol of the manufacturer, RNA was extracted from cells using the TRIzol reagent (Takara) and reverse transcribed using the PrimeScript RT reagent Kit (Takara). Quantitative RT-PCR was used to determine gene expression using the SYBR Green Realtime Master Mix (Takara). All values were normalized to the level of β-actin mRNA, and relative expression was calculated using the comparative cycle threshold ($2^{-\Delta\Delta CT}$) method.

The cells were harvested and washed twice with PBS before being lysed with RIPA. After 15 minutes of centrifugation at 13,000 rpm, the supernatant fraction was collected. The BCA Protein Assay Kit was used to assess the protein concentration in supernatants (Beyotime Biotechnology, Shanghai, China). Equivalent quantities of each protein sample were electrophoresed on SDS-PAGE gels and transferred to PVDF membranes (Pall Corporation). The primary antibodies directed against the following proteins were: ISG15, PRV-gE, phospho-STAT1 (Tyr701), STAT1, phospho-STAT2 (Tyr690), STAT2, IRF9 (1:3000 dilution; Cell Signaling). Secondary antibodies conjugated with horseradish peroxidase against rabbit or mouse (1:5000 dilution; Santa Cruz) were used. The ECL Western blotting Analysis System was used to reveal protein bands (Tanon, Shanghai, China). Densitometry was performed with ImageJ software and standardized against β-actin.

**Immunofluorescence assay.** PK15 cells were plated into a confocal dish and transfected with HA-STAT1 or myc-STAT2 plasmid. 4% paraformaldehyde was used to fix the monolayer cells, and 0.5% Triton X-100 were used to permeabilized at 4°C with (Solarbio Life Science, Beijing, China). Following a wash with PBS, cells were permeabilized in blocking solution (5% bovine serum albumin in PBS) for 1 h. Fixed cells were treated with a primary antibody specific for PRV-gE followed by an Alexa Fluor 488-conjugated secondary antibody against mouse (Proteintech). 4', 6-diamidino-2-phenylindole (DAPI) was used to stain the cell nuclei (Solarbio). Fluorescence pictures were acquired by confocal laser scanning microscopy (Nikon).

**ISG15 mutant plasmid.** The nonconjugative ISG15 plasmid pCAGGS-ISG15AA was constructed from pCAGGS-ISG15 using the site-directed mutagenesis kit (Beyotime), with the following primer pair: forward, 5'- TATA TGAATC TGCGCCTGCGGGCGGCCGGGACA GGG-3', and reverse, 5'- CCCTGTCCCGGCCGCCCGCAGGCGCAGATTCATATA-3'.

**siRNA silencing.** Twenty-four hours prior to transfection, PK15 cells were plated in 24-well plates. The cells were pretreated with IFNα for 12 h, and then transfected with control small interfering RNAs (siRNAs), or specific siRNAs against UbE1L using with 1 μL Lipofectamine RNAiMAX reagent (Invitrogen) per well. At 12 hpt, the cells were infected with PRV (MOI = 1). At 24 hpi, the culture media was changed with fresh medium containing 1000 U/mL IFNα, which was maintained throughout the infection duration. At 24 hpi, supernatants were collected for the viral titration, and cells were extracted for Western blotting and RT-qPCR analysis. The siRNAs sequences employed in this study were as follows: UbE1L no. 1: GCACUUCCCACCUGAUAAA; UbE1L no. 2: CAGCC UCACUCUUCAUGAU.

**Reporter gene assay.** Co-transfection of PK15 and ISG15$^{-/-}$ cells with the identified plasmid and the IFN-β-Luc or ISRE-Luc reporter plasmid (100 ng) plus the internal control pRL-TK reporter plasmid was performed (5 ng). Cells were treated with IFNα (1000 U/mL) for 12 h and were harvested to conduct dual-luciferase reporter assay (Promega). Firefly luciferase activity values were normalized to Renilla luciferase activity, and the relative fold changes in IFN-treated samples compared to IFN-untreated control were calculated.

**Co-immunoprecipitation (Co-IP) assays.** PK15 and ISG15$^{-/-}$ cells were treated with IFNα for 12 h, and then PK15 cells were transfected with siISG15 or ISG15$^{-/-}$ cells were transfected with ISG15AA. After 12 hpt, the cells were infected with PRV for 24 h, and the cell lysate was cleared by centrifugation at 14,000×g for 5 min at 4°C. Primary antibodies against HA, STAT1 or STAT2 (dilution 1:1000; Proteintech) were added to the supernatants. After three washes with TBS, SDS-PAGE sample buffer was added, and proteins were separated by SDS-PAGE and immunoblotted to determine STAT1 and STAT2 interaction. IB analysis of immunoprecipitations of ISG15$^{-/-}$ cells co-transfected with HA-ISG15AA, HA-ISG15 or Myc-STAT2/STAT1 expression plasmids. After 12 hpt, the cells were treated with IFNα for 24 h. Antibodies against Myc (dilution 1:1000; Proteintech) were added to the supernatants. Immunoprecipitation (IP) with Protein A+G Magnetic Beads (Beyotime) was done following the manufacturer's instructions. The IP samples with antibody against Myc were subjected to Western blotting with HA/Myc antibody.

**PRV challenge assay *in vivo*.** The ISG15 knockout mice were generated from the Cyagen Biosciences (Cyagen, China). The seven-week-old male ISG15$^{-/-}$ mice and WT mice were randomly divided into two groups consisting of 15 mice each, respectively. The mice had free access to pelleted food and water during the experimental period. Each mouse was intraperitoneally infected with PRV ($5\times10^3$ TCID$_{50}$) or PBS as control. The clinical symptoms, body weight and mortality were monitored daily. The brain and lung tissues were excised to detect the viral copies by absolute quantification real-time PCR, respectively. Blood serum were also collected and kept at 4°C to detect inflammatory factor through specific antibodies by

enzyme-linked immunosorbent assay (ELISA). In parallel, the brain tissues were fixed in neutral-buffered formalin for histological analysis. All the animal experiments used in this study were approved by the Animal Ethics Committee of Henan Agricultural University.

**Statistical analysis.** GraphPad Prism 8 software was used to conduct statistical comparisons. The difference between groups was determined using Student's *t*-tests, and P values less than 0.05 were considered statistically significant ($p < 0.05$). The standard errors of the mean (SEM) of at least three independent experiments are shown for each data.

## Supporting information

**S1 Fig. ISG15 inhibits PRV replication in MEF cells.** (A to C) MEF cells were transfected with empty-vector, or a plasmid expression expressing ISG15, or control siRNA or ISG15 siRNA, and then infected with PRV (MOI = 1) for 24 h. The mRNA and protein expression of PRV-gE, and PRV titer were detected by RT-qPCR, Western blot and plaque assays respectively. Each experiment was repeated at least three times separately. *, $p < 0.05$; **, $p < 0.01$; ***, $p < 0.001$ (*t*-test).
(TIF)

**S2 Fig. Antiviral activity of ISG15 against PRV relies on free ISG15.** (A and B) MEF cells were transfected with UbE1L siRNA, or ISG15AA, or ISG15, and then were infected with PRV. The expression of PRV-gE and ISG15 and PRV titer were detected by Western blot and plaque assays, respectively. **, $p < 0.01$ (*t*-test).
(TIF)

## Author Contributions

**Conceptualization:** Huimin Liu, Chen Li, Wenfeng He, Guoqing Yang, Lu Chen, Hongtao Chang.

**Data curation:** Huimin Liu, Chen Li, Wenfeng He.

**Funding acquisition:** Huimin Liu, Lu Chen.

**Investigation:** Chen Li, Wenfeng He, Jing Chen.

**Methodology:** Huimin Liu, Chen Li, Wenfeng He.

**Supervision:** Lu Chen, Hongtao Chang.

**Writing – original draft:** Huimin Liu, Chen Li.

**Writing – review & editing:** Huimin Liu, Lu Chen, Hongtao Chang.

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
