## [Decision Letter · Decision Letter 0]

29 Aug 2022

Dear Pro Liu,

Thank you very much for submitting your manuscript "Free ISG15 inhibits Pseudorabies virus infection by positively regulating type I IFN signaling" for consideration at PLOS Pathogens. As with all papers reviewed by the journal, your manuscript was reviewed by members of the editorial board and by several independent reviewers. In light of the reviews (below this email), we would like to invite the resubmission of a significantly-revised version that takes into account the reviewers' comments.

Dear Dr. Liu,

Your manuscript was reviewed by three expert virologists. All noted that the work is highly significant and of high interest to the general virology community. However, they did point out some significant concerns that should be addressed. If you are able to provide the additional proposed experimentation in a timely fashion, then we would like to reconsider this manuscript.

Cheers,

Eain Murphy

We cannot make any decision about publication until we have seen the revised manuscript and your response to the reviewers' comments. Your revised manuscript is also likely to be sent to reviewers for further evaluation.

Sincerely,

Eain A Murphy, Ph.D.

Associate Editor

PLOS Pathogens

Shou-Jiang Gao

Section Editor

PLOS Pathogens

Kasturi Haldar

Editor-in-Chief

PLOS Pathogens

orcid.org/0000-0001-5065-158X

Michael Malim

Editor-in-Chief

PLOS Pathogens

orcid.org/0000-0002-7699-2064

Dear Dr. Liu,

Your manuscript was reviewed by three expert virologists. All noted that the work is highly significant and of high interest to the general virology community. However, they did point out some significant concerns that should be addressed. If you are able to provide the additional proposed experimentation in a timely fashion, then we would like to reconsider this manuscript.

Cheers,

Eain Murphy

Reviewer's Responses to Questions

**Part I - Summary**

Reviewer #1: In this manuscript, the authors present a molecular mechanism of ISG15-mediated antiviral response against pseudorabies virus. Viruses trigger robust IFN response in infected cells, and IFN via IFN-induced proteins, called ISGs, mediates the antiviral protection in infected and uninfected cells. In this study, the authors examined how ISG15, a potent antiviral ISG, exerts its antiviral effects using a novel, previously unknown mechanism. ISG15 is known to function by conjugating with target proteins; however, the current study suggests that ISG15 can also function by a non-conjugation mechanism. Pseudorabies infection caused robust induction of ISG15, tested by complementary approaches, in PK15 cells. ISG15 KO cells showed enhanced viral replication and reduced IFN production. Using conjugation-defective ISG15 mutant, the authors elegantly showed that the antiviral function of ISG15 was not dependent on conjugated proteins. Finally, they showed that ISG15 promotes IFN-signaling by supporting STAT1-STAT2 complex formation and nuclear translocation. The physiological relevance of the study was examined in ISG15 KO mice, which were more susceptible to virus infection due to increased viral load and neuronal and pulmonary inflammation. Overall, the study is well-designed and uses complementary approaches and appropriate controls. Some weaknesses, as pointed out below, may help improve the study.

Reviewer #2: In this manuscript, Liu et al report that ISG15 expression is functionally linked to PRV replication. They found that ISG15 induction by PRV is MOI and time dependent as measured by Western blot, IF ad qPCR analyses. Of note, ISG15 knockout led to increased replication of PRV (about 2to 3-fold). This was attributed to the activity of free ISG15 as opposed to the conjugated one. In a mouse infection model, ISG15 deficiency was mirrored by more severe pathology and rapid death upon challenge with PRV. This in vivo phenotype is a strength of described work. There are also two observations on ISG15 connected to type I IFN production and STAT signaling mediated by IFN. While such results are intriguing, large gaps exist. This is quite evident, especially at the mechanical level where additional evidence is required. Overall, this study suggests a potential relation of ISG15 and PRV replication. Major concerns need to be addressed, which include, missing links, marginal effects of ISG15 (in infected cells), relevance of MOI, gE and data interpretation.

Reviewer #3: The manuscript by Liu et al. investigates the mechanism by which ISG15 inhibits the alphaherpesvirus, pseudorabies virus (PRV), infection. The authors previously demonstrated that PRV infection increased ISG15 expression and that its overexpression was sufficient to reduce viral replication. However, the mechanism by which ISG15 exerted antiviral activity remained unexplored. Here, using a combination of in vitro and in vivo PRV infections the authors argue that the ISG15 monomer (i.e., unconjugated) restricts PRV infection through its role in facilitating phosphorylation of STAT1 and STAT2. Overall, the manuscript addresses an interesting and important question, that is, what the mechanism of ISG15 antiviral activity during PRV infection is. However, the manuscript falls short from supporting this conclusion. In its current state, there are several severe deficits that make interpretations of its data difficult, including a lack of important controls and quantifications, and general difficulties in reading. Below are several suggestions that will help strengthen the manuscript.

**Part II – Major Issues: Key Experiments Required for Acceptance**

Reviewer #1: - The entire study was performed in one cell line, PK15, although it is unclear which cells were used for the ISG15 KO studies. The figure legends only show Wt and KO cells, but no details are provided.

- The pathogenesis study was done in mice without testing whether the phenomenon or the mechanism would be valid in the mouse system. Is the conjugation-independent ISG15 mechanism functional in mouse cells as well?

- There is a disconnect between the role of ISG15 in IFN production and IFN signaling. The IFN-signaling part has been followed up by testing STAT1-STAT2 complex formation. However, how ISG15 functions in the IFN production pathway is not known.

- For the mechanistic studies, it is unclear whether ISG15 interacts with either STAT1 or STAT2 or both. How unconjugated ISG15 promotes complex formation was not studied. Does ISG15AA interact with the STATs?

- Fig 6C-E, it is unclear which blots indicate pY-STAT; please label appropriately. Also, the nuclear translocation image is not very convincing in Fig 7D.

Reviewer #2: Point to be addressed:

1) As it stands, Figure 1 shows the expression of ISG15 in response to PRV infection. While there is a dynamic change, it is less clear why the level of ISG15 is reduced panel B). If gE is responsible, what does it do? Additional work will be required. Also, data in panel F seem to be hard to interpret.

2) ISG15 marginally reduced PRV replication (about 2-fold) as measured by plaque assay (Fig 2A). What is the limit of detection? There is no information on if data is from single or multiple experiments, which also appears a problem throughout the manuscript. If ISG15 inhibits viral release (Fig. 2E), how does it occur?

3) Figure 4 indicates a role of ISG15 in the induction of type IFN by PRV. Is it a direct or indirect effect? Something is missing.

4) Figure 5 shows a role of ISG15 in the antiviral effect mediated type I IFN in the range of 0.5 to1-fold (panel B and C). Again, it is unclear what is the limit of detection. Figure 6 shows a requirement of ISG15 in the STAT1/STAT2 interaction. Does ISG15 serve as a scaffold? Additional data will be helpful.

5) Experimental data in the manuscript are solely derived from a single cell type. Corroboration with other cells will strengthen the authors’ conclusions. Moreover, the gE null virus should be included for viral growth and mouse infection analyses. The manuscript is not clearly written, with typos, errors, and confusing statements (for example, lines 10-11, pp; lines 7-13, pp6).

Reviewer #3: Figure 1D: Use of gE has a marker for assessing neighboring cell infection is not useful as UV inactivation results in no detectable gE expression. Moreover, from the images it appears that most cells are infected with WT PRV, thus it is also not possible to tell whether ISG15 is induced in a neighboring cell.

Comments regarding gE being required for the induction of ISG15 are unwarranted. In the gE deleted virus there is still robust ISG15 expression.

Figure 2D a control should be added to show the complement of ISG1-/-

In the text, it is written that fig 2E supernatant has a 3-fold reduction which that graph does not appear to agree with.

Fig 3C should include the expression levels of UbeIL to show how efficient the UbeIL knockdown was

There are multiple typos and grammatical errors throughout the paper

Conclusions surrounding 7A are not justified as the STAT1 IP is less efficient. Thus, taking into account IP efficiency, it appears that the same amount complex is IP’d.

**Part III – Minor Issues: Editorial and Data Presentation Modifications**

Reviewer #1: None

Reviewer #2: (No Response)

Reviewer #3: (No Response)

PLOS authors have the option to publish the peer review history of their article (what does this mean?). If published, this will include your full peer review and any attached files.

Reviewer #1: No

Reviewer #2: No

Reviewer #3: No
---

## [Editor Report · Decision Letter 1]

7 Oct 2022

Dear Pro Liu,

We are pleased to inform you that your manuscript 'Free ISG15 inhibits Pseudorabies virus infection by positively regulating type I IFN signaling' has been provisionally accepted for publication in PLOS Pathogens.

Best regards,

Eain A Murphy, Ph.D.

Associate Editor

PLOS Pathogens

Shou-Jiang Gao

Section Editor

PLOS Pathogens

Kasturi Haldar

Editor-in-Chief

PLOS Pathogens

orcid.org/0000-0001-5065-158X

Michael Malim

Editor-in-Chief

PLOS Pathogens

orcid.org/0000-0002-7699-2064

Dear Dr. Liu,

Thank you for your resubmission. it has been reviewed by the editors and we have come to a decision of accept without additional review. Congratulations. This is in response to you taking the previous reviews seriously and adjusting the resubmission accordingly and including additional data where requested.

Cheers,

Eain Murphy
---

## [Editor Report · Acceptance letter]

24 Oct 2022

Dear Pro Liu,

We are delighted to inform you that your manuscript, "Free ISG15 inhibits Pseudorabies virus infection by positively regulating type I IFN signaling," has been formally accepted for publication in PLOS Pathogens.

Best regards,

Kasturi Haldar

Editor-in-Chief

PLOS Pathogens

orcid.org/0000-0001-5065-158X

Michael Malim

Editor-in-Chief

PLOS Pathogens

orcid.org/0000-0002-7699-2064